# Physical Fitness and the Level of Technical and Tactical Training of Kickboxers

**DOI:** 10.3390/ijerph18063088

**Published:** 2021-03-17

**Authors:** Łukasz Rydzik, Tadeusz Ambroży

**Affiliations:** Institute of Sports Sciences, University of Physical Education in Krakow, 31-541 Kraków, Poland; tadek@ambrozy.pl

**Keywords:** physical fitness, technical and tactical indicator, kickboxing

## Abstract

Background: Kickboxing is a dynamically progressing combat sport based on various techniques of punches and kicks. The high level of physical fitness underlies the optimal development of technique in the competitors. The objective of this study was the assessment of the level of fitness of kickboxers and the relationships between fitness and technical and tactical training. Methods: The study included 20 kickboxers aged 18–32 demonstrating the highest level of sporting performance. Their body mass ranged from 75 to 92 kg and their height from 175 to 187 cm. The selection of the group was intentional, and the criteria included training experience and the sports level assessed by the observation of the authors and opinion of the coach. The level of fitness was evaluated with the use of selected trials of International Committee on the Standardization of Physical Fitness Tests and Eurofit tests. Aerobic capacity was tested and indicators of efficiency, activeness and effectiveness of attacks were calculated. Results: A significant correlation between the indicators of technical and tactical training and results of fitness tests was shown. Conclusions: There exists a correlation between efficiency, activeness and effectiveness of attacks and the speed of upper limbs, explosive strength, static strength of a hand, agility, VO_2_max and abdominal muscle strength.

## 1. Introduction

General physical fitness is the locomotor basis on which a competitor can develop their professional techniques. There are a variety of different techniques in kickboxing which can be combined [1]. However, a proper level of technical and tactical training is the most important element of a competitor’s success. Technical and tactical actions allow effective control in a bout and can almost entirely avoid a rival’s attacks, simultaneously using offensive actions (counterattacks) [2]. Timing plays a key role, as it allows conducting an effective attack while simultaneously avoiding a rival’s offensive actions [3]. Kickboxing is a combat sport in which competitors fight each other using kicks and punches [4]. Amateur fighters use protectors, which reduce trauma occurrence in fights [5]. There are many types of kickboxing (point fighting, light contact, kicklight, full contact, low kick) that have different rules. There are also many kickboxing organizations. The World Association of Kickboxing Organizations (WAKO) is the largest and the most significant of them [5]. 

Proper functioning of the cardiovascular system is the basis of the physical fitness of a kickboxer [6], which allows repeating highly intense actions during a whole fight, mainly because of the increase in the regeneration process [7]. Mean values of VO_2_max of elite male kickboxers found in the literature range from 54 to 69 mL/kg/min [8,9]. 

Combat sports characterized by great intensity of actions are mostly based on anaerobic sources because decisive technical actions depend on quick and strong moves [10]. The energetic system adenosine triphosphate (ATP) and phosphocreatine (PCr) is very important for kickboxers because a proper strong blow can cause termination of a fight ahead of time (knock-out) [4,6]. A basic energetic source for this type of action includes anaerobic glycolysis; aerobic sources are important at the end of a fight (optimal aerobic and anaerobic endurance is necessary). A competitor’s training should then include anaerobic power (dynamic kicks and punches) and strength and speed of upper limbs (blows and their combinations in attacks, blocks and ducks in defense) [11].

The strength of the muscles of upper and lower limbs plays an important role in winning in a kickboxing fight [12]. The results of isometric strength (e.g., the grip strength) are greatly accepted as indicators of the level of a kickboxer’s strength [13,14]. The training process of kickboxers is diversified both in the context of the intensity of the training and the necessity of developing a wide range of motor skills [8,15]. Sports training in kickboxing is a subject of interest of many researchers [5,16,17]. A kickboxing fight is acyclic and its conditions change often (coordination and agility conditions). It has a holistic impact on the trainees and uses the whole organism, getting all groups of muscles active. The constantly changing situation in a bout requires good coordination and an immediate response to rival’s actions. A bout duration is usually 3 × 2 min. and it characterizes many changes in effort intensity [11]. Physical effort is based on submaximal and maximal loads. The physiological profiles of competitors show that the physical training in kickboxing should be aimed at increasing both aerobic and anaerobic capacity. Due to training and starting loads (often at the level >90% VO_2_max), muscle glycogen becomes the main source of energy. After terminating effort due to fatigue, glycogene is almost entirely used [18]. Using muscle glycogene in a given muscle group depends on the dynamics of movement in the ring, the frequency of changes in the intensity of the effort, methods of throwing kicks and punches and defensive reactions based mostly on anaerobic changes. Restoring glycogene takes place in after-effort restitution and its rate depends on many factors since there are moments of working on lower levels of VO_2_max in competitions.

The optimal level of physical fitness of a competitor is the key element of efficiency in a sports competition. Thanks to defining the level of physical fitness, one can select training loads in the appropriate amounts of exercises with respect to both quality and quantity. Regular measurements of this level also allow the assessment of the effects of the training [19]. The strength and dynamics of upper limbs in kickboxing have been evaluated by measuring the distance in throwing a medicine ball [20,21], and the strength and the dynamics of lower limbs were evaluated by measuring the distance of a jump [8,22]. Kickboxers were observed to have high levels of strength, power, aerobic and anaerobic capacity combined with technical and tactical skills. This is the reason why physical training should be based on improving strength and capacity of the muscles in the limbs [23].

Due to detailed technical and tactical analyses it is possible to define competitors’ training indicators as well as prove the existence of a relationship between the level of training and physical fitness of the participants. Technical and tactical analyses are common methods used in modifying the process of sport training in a group of martial arts and combat sports coaches and competitors. Interesting articles on this topic can be found in judo [24,25,26].

The results of the analysis of selected literature show that studies are concerned with success prognosis based on morpho-functional, physiological, biomechanical and psychosomatic indices [24], as well as assessment of capacity during fights [25,26]. Other studies were concerned with movement analysis [8], traumas and starting consequences [5,16]. A considerable deficit of texts regarding the level of training and the physical fitness of competitors was noted. The main objective of this paper is the assessment of the level of physical fitness of kickboxers in the highest sport level as well as finding a relationship between the fitness level and the indicators of technical and tactical training. Finding this correlation will determine whether the level of fitness influences activeness, effectiveness and efficiency of attacks and whether it allows more effective planning of sport training. 

## 2. Materials and Methods

This study included 20 kickboxers presenting the highest level of sport. The selection of the group was intentional, and the criteria included training experience and the sports level assessed by observation of the authors and opinion of the coach. The participants were from 18 to 32 years old, their body mass ranged 75 to 92 kg and their height was between 175 and 187 cm. BMI of the participants ranged 24.13 to 28.73 kg/m^2^(Table 1). 

### 2.1. Physical Fitness Tests 

The physical fitness of the participants was assessed by selected tests taken from the tests developed by the International Committee on the Standardization of Physical Fitness Tests (ICSPFT) and European Fitness Test (EUROFIT) [27]. The entire test included the following: Aerobic capacity test—VO_2_max (description of the test below)Tapping—Assessment of speed of upper hands. The subject stands in front of a table with their feet spread and puts their worst hand on a rectangular pad. Their better hand is placed on a farther disc. They should touch both discs alternatively as quick as possible. The subject makes a total of 50 moves, they touch each disc 25 times. They take two tests and the best one is noted; the time is rounded to a decimal place.Standing long jump—Jumping with both feet from standing. The test measures the distance jumped in cm, which is an indicator of the possibility to quickly create strength. The subject stands with their feet lightly spread behind the start line, they bends their knees moving their arms backward, then they move their arms forward, bounce their feet from the ground and make a jump as long as possible. They land on both feet in a standing position. The test is taken twice. The longer jump is recorded, rounded to the nearest cm.Grip strength using a dynamometer. Evaluating the isometric strength. The subject has their feet lightly spread, the dynamometer lies close to fingers, arm down along the body but without touching the body. Short grip on a dynamometer using maximum strength, second arm loose along the body. Best of two tests is recorded; the result is rounded to 1 kg.Shuttle run (10 × 5 m). The subject runs on a signal to the second line 5 m away, crosses it with both feet and comes back. They run 10 times for a distance of 5 m, the time of the shuttle run is measured and rounded to a decimal place of a second.Pull-ups—Evaluating shoulder girdle strength counting the number of repetitions. The subject catches a bar, their hands are spread in line with their shoulders and they do an overhang. On a signal they bend their arms and pull up their body so their beard should be above the bar. After a moment of rest they return to an overhang. They repeat the exercise as many times as possible. The result is the number of repetitions.Sit-ups—Evaluating abdominal muscle strength. The subject lies on a mattress, their feet are 30 cm apart and their knees bent at 90 degrees, with hands on their neck. A partner holds the subject’s feet so they stay on the ground. On a signal the subject performs sit-ups touching their knees with their elbows and coming back to lying down. The test lasts 30 s.Flexibility test—The subject bends their torso forward when sitting down and the range of motion behind feet is measured in cm. The sitting subject moves a ruler with their hands on a box with a scale. The best of two tests is recorded.Cooper’s test—Running endurance—12-min run, distance is measured.

The tests were done by the authors, with tests 1–4 on the first day, and tests 5–9 on the second. The volume of training was reduced to 30–40% two days before the tests.

### 2.2. Measuring the Indicators of Technical and Tactical Training 

The analysis of a sports bout was done based on digital recording of a fight. Then, the indicators of technical and tactical training were computed using the following formulas [5].

Efficiency of the attack (S_a_)
$$\mathrm{Sa}\;=\;\frac{n}{N}$$

n—Number of attacks awarded 1 pt.^*^

* In K1 formula each fair hit is awarded 1 pt.

N—Number of bouts.

Effectiveness of the attack (E_a_)
$$\mathrm{Ea}\;=\;\frac{number\;of\;efective\;attacks}{number\;of\;all\;attacks}\;\times \;100$$

* An effective attack is a technical action awarded a point.

* Number of all attacks is the number of all offensive actions. 

Activeness of the attack (A_a_)
$$\mathrm{Aa}\;=\;\frac{number\;of\;all\;registered\;offensive\;actions\;of\;a\;kickboxer}{number\;of\;bouts\;fought\;by\;a\;kickboxer}$$

### 2.3. VO_2_max Measurement

The test of maximal oxygen intake (VO_2_max) was done with the use of the Margaria test. The participants climbed a step 40 cm tall. In the first 6-minute period the frequency of climbing was 15/min, in the second was 25/min. During both parts, heart rate was measured with sportster (Polar). The maximal oxygen intake was computed based on the formula in [28].
$${\mathrm{VO}}_{2}\max=\frac{\mathrm{HRmax}\left(\mathrm{VO}2\mathrm{II}-\mathrm{VO}2I\right)+\mathrm{HRII}*\mathrm{VO}2I*\mathrm{VO}2\mathrm{II}}{\mathrm{HRII}-\mathrm{HRI}}$$
where:

HRmax—max heart rate [beats/min.]

*HRmax computed according to Tanaka 2001 (208 − 0.7*age) [29]

HRI—heart rate during I part [beats/min.]

HRII—heart rate during II part [beats/min.]

VO2I—estimated oxygen intake during I part [mL/O /kg/min],

that requires ca. 22.0 [mL/O /kg/min]

VO2II—estimated oxygen intake during II part [mL/O /kg/min],

that requires ca. 23.4 [mL/O /kg/min]

### 2.4. Bioethical Committee 

Prior to participation in the tests, the competitors were informed about the research procedures, which were in accordance with the ethical principles of the Declaration of Helsinki WMADH (2000). Obtaining the competitors’ written consent was the condition for their participation in the project. The research was approved by the Bioethics Committee at the Regional Medical Chamber (No. 287/KBL/OIL/2020).

### 2.5. Statistical Analysis 

Statistical analysis of the data was done with the use of Statistica 13.1 by StatSoft. Parametric tests were used due to meeting the basic assumptions concerning the consistency of studied distributions to a normal distribution and the homogeneity of the variance. The consistency of the distributions to a normal distribution was evaluated with the use of a Shapiro–Wilk test, and the homogeneity of variance was evaluated with the use of a Levene test. All descriptive statistics (mean, median, minimum, maximum, 95% confidence intervals, 1st and 3rd quartile and standard deviation) were computed for all variables. The correlation of two variables of a normal distribution was evaluated with the use of a Pearson’s linear correlation coefficient. The level of statistical significance was set to *p* < 0.05.

## 3. Results

The results of the fitness tests of the participants are shown in Table 2. 

The mean level of aerobic capacity was 47.65 mL/kg/min and the results ranged from 41 to 56 (Table 3).

Activeness of the attack was 96.8 on average and it ranged from 64 to 133. Effectiveness of the attack was 47.85 on average and it ranged from 40.6 to 56.32. Efficiency of the attack was 50.45 on average and it ranged from 45 to 56 (Table 3 and Table 4). 

A strong negative correlation between aerobic capacity and the speed of upper limb and between aerobic capacity and shuttle run was shown as well as a strong positive correlation between aerobic capacity and standing long jump and between aerobic capacity and endurance. There was also a strong correlation between VO_2_max and static strength of both hands and between VO_2_max and abdominal muscle strength. It was also proven that body mass was strongly positively correlated with the speed of upper limbs and shuttle run as well as being negatively correlated with standing long jump and endurance. Participants who were quicker and more agile also had higher levels of the indicators of activeness, effectiveness and efficiency of attacks (strong negative correlation of the indicators with speed and shuttle run and strong positive correlation with standing long jump and endurance). Participants who had higher results of standing long jump or Cooper’s test also had higher levels of the indicators. The efficiency of attacks was correlated with abdominal muscle strength (Table 5). 

Participants who had higher levels of VO_2_max also had higher levels of indicators of activeness, effectiveness and efficiency of attacks. All correlations were strong and statistically significant. Moreover, lighter and shorter participants also had higher levels of the indicators. Correlations between body mass and the levels of indicators of activeness, effectiveness and efficiency of attacks were strong and the correlations between the height and the levels of indicators of activeness, effectiveness and efficiency of attacks were medium; all correlations were significant. Effectiveness of the attack was significantly negatively correlated to the BMI of the participants. The correlation had a medium strength (Table 6). 

## 4. Discussion

A kickboxing bout in a K1 rules competition is dynamic as well as comprehensive in the technical and tactical aspects [4]. Contenders who fight in the highest level competitions must have proper aerobic capacity. In this study, the mean result of the participants’ level of VO_2_max was 47.65 mL/kg/min, which can be interpreted as a high level of aerobic capacity [16,17]. In other combat sports the mean level of competitors’ VO_2_max was as follows: 40.8 mL/kg/min (judokas), 50.3 mL/kg/min (boxers), 58.4 mL/kg/min (MMA fighters) [30,31,32]. The participants of this study had better VO_2_max level than judokas, but were worse than MMA fighters and comparable to boxers. Statistical analysis showed a strong and significant correlation between the level of VO_2_max and the activeness, the effectiveness and the efficiency of the participants. This can show that the level of indicators of technical and tactical training depends on aerobic capacity and they can be related to general endurance of the organism (VO_2_max underlies the endurance) that is considered as the basis of physical possibilities of a competitor [33]. Statistical analysis also showed significant negative correlations between the speed of upper limbs and the indicators of technical and tactical training. The correlations were strong (with the efficiency of the attack) and medium (with other indices). Similarly, there were strong (with the efficiency) and medium (with other indices) correlations between shuttle run and the indicators. Thus it follows that the competitors whose results in speed and agility test were worse, were more active and had greater effectiveness and efficiency of their attacks than the participants with better results in those tests. The higher the speed of the upper limbs, the higher the number of competitor’s actions involving hand techniques in a round. Good upper limb speed also corresponds to better defensive actions. Due to great similarity in the actions involving upper limbs in both boxers and kickboxers, the analysis of the results of this study could be also done in the study of boxers and the results would be similar [34,35]. Competitors who had quicker upper limbs were able to use more techniques which resulted in increasing their activeness, effectiveness and efficiency indicators. Similar results were reached in assessing agility (as speed and coordination) which is characteristic and significant in kickboxing competitions. Thanks to a high level of agility it is possible to move more smoothly in a ring, which makes attacks more effective (one can surprise a rival with feints, change of pace and anticipating the attack) and improves defensive actions (dodging, ducks, turns). Speed and coordination are basic elements of a kickboxing fight and they underlie proper timing, which means using a technique in the right moment [8,15,23]. 

There were strong positive correlations between the results of most of the fitness tests and the levels of the indicators of technical and tactical training. It was shown that a high level of maximum anaerobic power was assessed with the use of the standing long jump test. It is worth noticing that Ambroży et al. proved that a high round kick was the most effective lower limb technique, and could often end a fight with a knock-out [4]. Dynamic strength determines the efficiency of doing kicks and it could also improve the indicators of technical and tactical training [36]. Static strength is an important element of the motor preparation of a competitor. Its high level gives the possibility of increasing the technical potential of a kickboxer [37]. In this study we showed the existence of medium strength significant correlations between the technical and tactical training indicators and the static strength of a hand. The analysis of the results shows that there is a medium strength relationship between the abdominal muscle strength and the effectiveness of a competitor in a kickboxing fight. The training process in kickboxing is based on comprehensive development of abdominal muscles, which guarantees an effective defense, protecting a fighter’s torso. This is the reason why this relationship can be a direct effect of the training methods used. 

This study showed a negative correlation between body mass and height vs. the level of indicators of technical and tactical training. The strength of the correlations was usually medium, and only in the case of the relationship between body mass and activeness of the attack was the correlation strong. Participants competing in lower weight categories can punch and kick faster but at the cost of the strength of a blow [38]. That could be a reason why lighter and shorter participants had better results of activeness, but also effectiveness and efficiency of attacks in comparison to heavier and taller kickboxers. Tests conducted in this study did not prove the existence of significant relationships between shoulder girdle strength (pull-ups) and the level of the indicators of technical and tactical training. Thus it follows that shoulder girdle strength is not a significant element of a kickboxing competition according to K1 rules. Similarly, there was no significant relationship between the level of agility and the level of the indicators of technical and tactical training. Competitors fighting according to K1 rules use mostly low kicks on the thigh or high round kicks that do not require an above average developed agility level. 

This way of fighting could point, for example, to the lack of agility predispositions in some competitors, which, as can be seen, is not an element that could decide the win in a kickboxing fight. 

## 5. Conclusions

Activeness, effectiveness and efficiency of the competitors expressed by the indicators of technical and tactical training show a strong correlation to the level of maximum oxygen intake VO_2_max. It follows that kickboxers should work out the optimal level of aerobic capacity in a preparation term and then maintain this during the competitions. It should impact their starting possibilities. 

The level of the speed of upper limbs and agility influence the starting possibilities measured with the use of the indicators of technical and tactical preparations. This is closely connected to efficiency of a kickboxing fight. 

Efficiency, effectiveness and activeness of an attack depend on the level of muscle strength of upper, middle and lower parts of the body. 

Somatic features of the competitors influence activeness, effectiveness and efficiency of attacks. The relationships show the necessity of controlling body mass before the start of a competition and keeping it at the optimum level in the aspect of weight categories. 

### Practical Implication 

The training process of kickboxers fighting according to K1 rules should be based on the comprehensive development of a competitor in the aspects of strength, speed and endurance, while keeping the optimal body weight should underlie the training process. 

## Figures and Tables

**Table 1 ijerph-18-03088-t001:** Anthropometric characteristic of the participants.

Variables	No	Mean	95% Confidence Interval		Median	Minimum	Maximum	1st Quartile	3rd Quartile	Standard Deviation
Body mass	20	84.90	82.59	87.21	85.50	75.00	92.00	83.00	88.50	4.93
Height	20	181.05	179.46	182.64	180.00	175.00	187.00	179.00	183.50	3.39
BMI	20	26.04	25.46	26.62	25.99	24.13	28.73	25.15	26.73	1.24

BMI-Body Mass Index.

**Table 2 ijerph-18-03088-t002:** The rezults of fitness test.

Variables	Number	Mean	95% Confidence Interval		Median	Minimum	Maximum	1st Quartile	3rd Quartile	Standard Dev.
Plate tapping [s]	20	7.64	7.17	8.10	7.25	6.46	9.43	6.89	8.18	1.00
Standing long jump [cm]	20	205.25	198.14	212.36	210.00	167.00	225.00	198.00	216.50	15.19
Cooper’s test [m]	20	3086.20	2928.53	3243.87	3003.50	2656.00	3920.00	2837.00	3327.50	336.88
Static strength of a right hand [kg]	20	55.96	55.06	56.85	56.16	51.22	58.65	55.06	57.13	1.91
Static strength of a left hand [kg]	20	54.70	53.67	55.73	55.12	50.26	58.30	53.22	56.44	2.20
Pull-ups on a bar [n]	20	18.05	16.21	19.89	17.00	10.00	26.00	15.50	21.50	3.94
Shuttle run [s]	20	11.02	10.62	11.42	10.93	10.01	13.45	10.36	11.38	0.85
Flexibility [cm]	20	15.98	15.67	16.29	15.90	15.00	18.00	15.65	16.30	0.65
Sit-ups [n]	20	30.35	28.03	32.67	31.50	23.00	39.00	25.50	34.50	4.97

**Table 3 ijerph-18-03088-t003:** VO2max.

Variables	Number	Mean	95% Confidence Interval		Median	Minimum	Maximum	1st Quartile	3rd Quartile	Standard Dev.
VO_2_max [mL/kg/min]	20	47.65	45.59	49.71	49.00	41.00	56.00	43.00	51.00	4.39

**Table 4 ijerph-18-03088-t004:** Activeness, effectiveness and efficiency of attacks.

Variables	Number	Mean	95% Confidence Interval		Median	Minimum	Maximum	1st Quartile	3rd Quartile	Standard Dev.
Activeness	20	96.80	89.46	104.14	92.00	64.00	133.00	89.00	102.50	15.69
Effectiveness	20	47.84	45.00	50.69	45.29	40.60	56.32	42.44	53.79	6.08
Efficiency	20	50.45	48.83	52.07	50.50	45.00	56.00	48.00	53.00	3.47

**Table 5 ijerph-18-03088-t005:** The influence of selected variables on the results of fitness tests.

Pearson’s Linear Correlation Coefficient r Level of Significance *p*	VO_2_max	Body Mass	Height	BMI	Activeness	Effectiveness	Efficiency
Plate tapping [s]	−0.89	0.80	0.52	0.55	−0.55	−0.79	−0.82
	0.001	0.001	0.020	0.013	0.013	0.001	0.001
Standing long jump [cm]	0.85	−0.72	−0.57	−0.40	0.52	0.74	0.85
	0.001	0.001	0.009	0.077	0.019	0.001	0.001
Cooper’s test [m]	0.87	−0.87	−0.59	−0.50	0.80	0.67	0.70
	0.001	0.001	0.007	0.026	0.001	0.001	0.001
Static strength of a right hand [kg]	0.74	−0.60	−0.61	−0.22	0.50	0.51	0.77
	0.001	0.005	0.004	0.350	0.026	0.021	0.001
Static strength of a left hand [kg]	0.67	−0.55	−0.42	−0.54	0.34	0.65	0.73
	0.001	0.012	0.065	0.015	0.143	0.002	0.001
Pull-ups on a bar [n]	−0.22	0.28	0.44	−0.07	−0.19	−0.19	−0.22
	0.349	0.238	0.052	0.766	0.430	0.415	0.349
Shuttle run [s]	−0.85	0.82	0.71	0.33	−0.63	−0.70	−0.85
	0.001	0.001	0.001	0.155	0.003	0.001	0.001
Flexibility [cm]	−0.14	−0.01	0.14	−0.03	−0.06	−0.06	−0.10
	0.550	0.970	0.561	0.903	0.805	0.817	0.666
Sit-ups [n]	0.52	−0.26	−0.28	−0.13	0.13	0.42	0.49
	0.019	0.263	0.234	0.587	0.587	0.068	0.027

**Table 6 ijerph-18-03088-t006:** The influence of selected variables on the activeness, the effectiveness and the efficiency of attacks.

Pearson’s Linear Correlation Coefficient (r) Level of Significance *p*	VO_2_max	Body Mass	Height	BMI
Activeness	0.72	−0.82	−0.58	−0.35
	0.001	0.001	0.007	0.131
Effectiveness	0.70	−0.74	−0.58	−0.51
	0.001	0.001	0.007	0.021
Efficiency	0.88	−0.71	−0.69	−0.32
	0.001	0.001	0.001	0.175

## Data Availability

The data presented in this study are available on request from the corresponding author.

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
