# Peer review of "Physical Fitness and the Level of Technical and Tactical Training of Kickboxers"

_ijerph, 2021, doi:10.3390/ijerph18063088_

Round 1
Reviewer 1 Report
The main result of the study is that kickboxers who are overall more trained and fit are likely more effective in kickboxing fights. Many other results are also very expectable and their theoretical implications rather vague. The originality of the study is therefore rather low in my opinion.
It would be more interesting to observe a point that is only mentioned briefly in the conclusion: The different VO2max of judoka, kickboxers, boxers and MMA fighters. Here, research into the causes would be very interesting to read.
Nevertheless, the study presented is methodologically and theoretically sound. Only minor typos occur.
Author Response
Dear Reviewer,
Thank you very much for your time and valuable comments. The detailed list of responses is given below.
Your Sincerely,
Rydzik- Coresponding author
The role of the study is to be applied in everyday work of kickboxing clubs, which was already done. The objective of the study was to justify the relationship between indicators of physical fitness and indicators of tactical and technical training. Using self-constructed indices of tactical and technical training based on similar indices used in judo, which are in common use, was one of the values of the study. Our next study will verify the usefulness of this type of indices in kickboxing. Additionally, thanks to our research, coaches learn about development of particular locomotor skills to improve competitor’s efficiency. There has not been any research in this area. Your suggestion about comparing VO2max between different types of combat sports is interesting and we will definitely use it in our research. We would only like to notice that this article was send to a special edition that concerns physical fitness and training effects in individual sports.
Reviewer 2 Report
Revision of the manuscript "Physical fitness and the level of technical and tactical training of kickboxers".
After reading the article, a number of improvements to the article have been made, which I recommend to the authors. Step by step:
Key words.
There are too many words that are not "key". For example: "training control; technical and tactical preparation; trainer control". I suggest authors to reduce similar words and look for the right one in a thesaurus.
Introduction section.
Although this section is well thought out, I suggest that the authors specify and detail the variables of the research. For example the definition of Kickboxing is reduced to "sport in which competitors fight each other using kicks and punches". Where is the sport most commonly practised?
Are there broader definitions?
Material and methods section.
The 20 participants... How were they chosen, how long were they tested, did they do the Alpha battery tests all at the same time, who measured the tests, and the gender of the participants?
Table 1 contains words that are not in English and are not understood.
This section needs to be profoundly improved.
Results section.
Tables 5 and 6 should not contain "p=" and "p<". Those annotations should appear in a "Note" at the bottom of the table.
Conclusions section.
I suggest to the authors that this section be placed in paragraph form.
Author Response
Dear Reviewer,
Thank you very much for your time and valuable comments, which all have been considered and incorporated. The detailed list of responses is given below. We hope that the modifications and explanation will be acceptable for you.
Yours sincerely,
Rydzik, corresponding author
Key words.
There are too many words that are not "key". For example: "training control; technical and tactical preparation; trainer control". I suggest authors to reduce similar words and look for the right one in a thesaurus.
A:key words training control, ; trainer control were removed, and technical and tactical preparation were changed to technical and tactical indicator
Introduction section.
Although this section is well thought out, I suggest that the authors specify and detail the variables of the research. For example the definition of Kickboxing is reduced to "sport in which competitors fight each other using kicks and punches". Where is the sport most commonly practised?
Are there broader definitions?
A: A broader definition of kickboxing was added
Material and methods section.
The 20 participants... How were they chosen, how long were they tested, did they do the Alpha battery tests all at the same time, who measured the tests, and the gender of the participants?
A: We added: The choice of the study group was purposeful, the criterion was the training experience and the sports level assessed by observation of the authors and an opinion of the coach. The order of the tests was added, on the first day there were tests 1 – 4, on the second day – tests 5 – 9. Two days before the study the intensity of the training was lowered to 30-40%.
Table 1 contains words that are not in English and are not understood.
This section needs to be profoundly improved.
A: Corrected
Results section.
Tables 5 and 6 should not contain "p=" and "p<". Those annotations should appear in a "Note" at the bottom of the table.
A:corrected
Conclusions section.
I suggest to the authors that this section be placed in paragraph form.
A:corrected
Reviewer 3 Report
General Comments
This manuscript has the potential to be of real interest; however, there are substantial revisions required throughout the manuscript, as detailed below. Please also check the use of English throughout.
In the introduction you need to explain the key physical demands of kickboxing, based on the movement demands, duration and intensity of the tasks, so that you can clearly identify which physical characteristics should be developed and evaluated.
A brief summary of the testing procedures for each method of evaluating physical performance must be presented, it should be possible for someone to replicate your study, based on the information you provide.
Specific Comments
Line 10: Change ‘paper’ to ‘study’
Line 11: Change ‘relation’ to ‘relationships’
Line 12: Please add subject characteristics
Line 15: It appears as though you are referring to multiple correlations and not a single correlation, so please clarify this. In addition, while you state that they were significant were they meaningful? Both the p values and the r values should be presented, and the strength of the association should be described. Please keep in mind that the magnitude of the correlation is arguably more important than the level of significance.
Line 18: The 2 in VO2max should be subscript.
Lines 27-30: Please re-phrase this section for clarity as it is rather vague as currently written.
Lines 45-46: This needs to be explored in more detail and re-written for clarity.
Lines 62-70: These need to be refined and explained in detail. For example, speed of upper limbs appears to be reported (in table 2) in seconds. Therefore, this is not speed, this is time to completion, in which a lower time indicates a greater speed. Explosive strength is simply standing long jump distance and not a measure of strength, but dynamic expression of force. Please see Winter et al. (2016). "Misuse of "Power" and other mechanical terms in Sport and Exercise Science Research." The Journal of Strength & Conditioning Research 30(1): 292-300 regarding the use of the term explosive in this context. The agility test is not an assessment of agility, as there is no reaction to a stimulus. Please check that each test actually evaluates what you claim is being evaluated and amend accordingly, both here and throughout the manuscript.
Lines 98-100: Please also include the 95% confidence intervals for each correlation and explain how the magnitude of each correlation was interpreted. In addition, due to the multiple correlations you must correct each p value for familywise error rates. This must them be updated throughout the results and discussion.
There is no point providing any further comments beyond this point as the results and discussion sections will need to be re-written, based on the amendments listed above.
General Comments
This manuscript has the potential to be of real interest; however, there are substantial revisions required throughout the manuscript, as detailed below. Please also check the use of English throughout.
In the introduction you need to explain the key physical demands of kickboxing, based on the movement demands, duration and intensity of the tasks, so that you can clearly identify which physical characteristics should be developed and evaluated.
A brief summary of the testing procedures for each method of evaluating physical performance must be presented, it should be possible for someone to replicate your study, based on the information you provide.
Specific Comments
Line 10: Change ‘paper’ to ‘study’
Line 11: Change ‘relation’ to ‘relationships’
Line 12: Please add subject characteristics
Line 15: It appears as though you are referring to multiple correlations and not a single correlation, so please clarify this. In addition, while you state that they were significant were they meaningful? Both the p values and the r values should be presented, and the strength of the association should be described. Please keep in mind that the magnitude of the correlation is arguably more important than the level of significance.
Line 18: The 2 in VO2max should be subscript.
Lines 27-30: Please re-phrase this section for clarity as it is rather vague as currently written.
Lines 45-46: This needs to be explored in more detail and re-written for clarity.
Lines 62-70: These need to be refined and explained in detail. For example, speed of upper limbs appears to be reported (in table 2) in seconds. Therefore, this is not speed, this is time to completion, in which a lower time indicates a greater speed. Explosive strength is simply standing long jump distance and not a measure of strength, but dynamic expression of force. Please see Winter et al. (2016). "Misuse of "Power" and other mechanical terms in Sport and Exercise Science Research." The Journal of Strength & Conditioning Research 30(1): 292-300 regarding the use of the term explosive in this context. The agility test is not an assessment of agility, as there is no reaction to a stimulus. Please check that each test actually evaluates what you claim is being evaluated and amend accordingly, both here and throughout the manuscript.
Lines 98-100: Please also include the 95% confidence intervals for each correlation and explain how the magnitude of each correlation was interpreted. In addition, due to the multiple correlations you must correct each p value for familywise error rates. This must them be updated throughout the results and discussion.
There is no point providing any further comments beyond this point as the results and discussion sections will need to be re-written, based on the amendments listed above.
Author Response
Dear Reviewer,
Thank you very much for your time and valuable comments, which all have been considered and incorporated. The detailed list of responses is given below. We hope that the modifications and explanation will be acceptable for you.
Yours sincerely,
Rydzik, corresponding author
In the introduction you need to explain the key physical demands of kickboxing, based on the movement demands, duration and intensity of the tasks, so that you can clearly identify which physical characteristics should be developed and evaluated.
A: Added, physical requirements of a kickboxing bout were shown through physiological and locomotor characteristic connected to specific fitness type. Based on this analysis one can show which virtues should be assessed and developed..
A brief summary of the testing procedures for each method of evaluating physical performance must be presented, it should be possible for someone to replicate your study, based on the information you provide.
A: Due to restricted volume of the article there was only a short description of the tests used. The detailed description can be found in the appropriate position in References.
Specific Comment
Line 10: Change ‘paper’ to ‘study’
Line 11: Change ‘relation’ to ‘relationships’
Line 12: Please add subject characteristics
A:Corrected
Line 15: It appears as though you are referring to multiple correlations and not a single correlation, so please clarify this. In addition, while you state that they were significant were they meaningful? Both the p values and the r values should be presented, and the strength of the association should be described. Please keep in mind that the magnitude of the correlation is arguably more important than the level of significance.
Line 18: The 2 in VO2max should be subscript.
A: Corrected
Lines 27-30: Please re-phrase this section for clarity as it is rather vague as currently written.
A: Re-phrased for clarity
Lines 45-46: This needs to be explored in more detail and re-written for clarity.
A:Re-written for clarity
Lines 62-70: These need to be refined and explained in detail. For example, speed of upper limbs appears to be reported (in table 2) in seconds. Therefore, this is not speed, this is time to completion, in which a lower time indicates a greater speed. Explosive strength is simply standing long jump distance and not a measure of strength, but dynamic expression of force. Please see Winter et al. (2016). "Misuse of "Power" and other mechanical terms in Sport and Exercise Science Research." The Journal of Strength & Conditioning Research 30(1): 292-300 regarding the use of the term explosive in this context. The agility test is not an assessment of agility, as there is no reaction to a stimulus. Please check that each test actually evaluates what you claim is being evaluated and amend accordingly, both here and throughout the manuscript.
A: Standard tests were used, widely described in literaturę.. We modified the desription of the tests explaining method of measurement.
Lines 98-100: Please also include the 95% confidence intervals for each correlation and explain how the magnitude of each correlation was interpreted. In addition, due to the multiple correlations you must correct each p value for familywise error rates. This must them be updated throughout the results and discussion.
A: the 95% confidence intervals for each correlation were added
There is no point providing any further comments beyond this point as the results and discussion sections will need to be re-written, based on the amendments listed above.
Round 2
Reviewer 2 Report
Well done. I recommend accept in present form.
Author Response
Dear Reviewer,
Thank you very much for you reviews.
Yours sincerly,
Rydzik-corresponding author
Reviewer 3 Report
General Comments
This manuscript has the potential to be of real interest; however, there are substantial revisions required throughout the manuscript, as detailed below. Please also check the use of English, including punctuation and grammar throughout, e.g., there are numerous occasions where apostrophes are used incorrectly and where word choice is poor.
Thank you for adding some additional information regarding the key physical demands of kickboxing; however, greater detail is still required.
A brief summary of the testing procedures for each method of evaluating physical performance must be presented, it should be possible for someone to replicate your study, based on the information you provide. In the response to this comment you stated that the restricted volume of the article permits only a short description, but there is no limit on the wordcount for this journal, so please provide the specific details to ensure that the reader could replicate your methods.
Specific Comments
Lines 39-42: This needs to be better contextualised, in terms of the energy and muscular requirements for the high intensity efforts and actions and the associated recovery between such repeated high intensity actions. This should reflect the demands on the energy systems and the requirements for rapid force production. If there is no specific information regarding this sport you can infer the interaction between the energy systems from other sports where repeated high intensity actions are performed, ideally using other combat sports.
Line 43: What is meant by ‘anaerobic strength’? Strength, by definition is anaerobic. Why then also repeat strength after the parentheses?
Lines 46-49: Please re-write this for clarity as it is not clear what you mean here.
Line 66: There should be a space between physiological and biomechanical
Line 81: Please provide a more appropriate title for the table, e.g. Physical or anthropometric characteristics of participants
Line 87: The definition of aerobic capacity should either be refined for accuracy or deleted.
Line 93: This does not assess strength, which is clear from the fact that the units of measurment is cm and therefore this is a measure of jump distance, which indicates the participants ability to rapidly produce force and project themselves during a horizontal jump. It is not a measure of strength.
Line 94: Change static to isometric
Line 106: The intensity or volume?
Lines 152: THIS WAS NOT ADDRESSED FROM THE PREVIOUS REVIEW (95%CI were added in relation to the mean values)- Please also include the 95% confidence intervals for each correlation and explain how the magnitude of each correlation was interpreted. In addition, due to the multiple correlations you must correct each p value for familywise error rates. This must them be updated throughout the results and discussion. Please also provide the threshold by which the r values are interoretted.
There is no point providing any further comments beyond this point as the results and discussion sections will need to be re-written, based on the amendments listed above.
Author Response
Dear Reviewer,
Thank you very much for your time and valuable comments, which all have been considered and incorporated. The detailed list of responses is given below. We hope that the modifications and explanation will be acceptable for you.
Yours sincerely,
Rydzik, corresponding author
This manuscript has the potential to be of real interest; however, there are substantial revisions required throughout the manuscript, as detailed below. Please also check the use of English, including punctuation and grammar throughout, e.g., there are numerous occasions where apostrophes are used incorrectly and where word choice is poor.
A: Punctuation and grammar has been corrected
Thank you for adding some additional information regarding the key physical demands of kickboxing; however, greater detail is still required.
A: Added more specific information
A brief summary of the testing procedures for each method of evaluating physical performance must be presented, it should be possible for someone to replicate your study, based on the information you provide. In the response to this comment you stated that the restricted volume of the article permits only a short description, but there is no limit on the wordcount for this journal, so please provide the specific details to ensure that the reader could replicate your methods.
A: Added detailed description of each test
Lines 39-42: This needs to be better contextualised, in terms of the energy and muscular requirements for the high intensity efforts and actions and the associated recovery between such repeated high intensity actions. This should reflect the demands on the energy systems and the requirements for rapid force production. If there is no specific information regarding this sport you can infer the interaction between the energy systems from other sports where repeated high intensity actions are performed, ideally using other combat sports.
A: Substantial corrections in the introduction were madein order to describe the parameters in a more detailed way Added more items in References
Line 43: What is meant by ‘anaerobic strength’? Strength, by definition is anaerobic. Why then also repeat strength after the parentheses?
A: Corrected
Lines 46-49: Please re-write this for clarity as it is not clear what you mean here.
A: Re-written
Line 66: There should be a space between physiological and biomechanical
A: corrected
Line 81: Please provide a more appropriate title for the table, e.g. Physical or anthropometric characteristics of participants
A: corrected
Line 87: The definition of aerobic capacity should either be refined for accuracy or deleted.
A: deleted
Line 93: This does not assess strength, which is clear from the fact that the units of measurment is cm and therefore this is a measure of jump distance, which indicates the participants ability to rapidly produce force and project themselves during a horizontal jump. It is not a measure of strength.
A: corrected
Line 94: Change static to isometric
A: changed
Line 106: The intensity or volume?
A; This is intensity
Lines 152: THIS WAS NOT ADDRESSED FROM THE PREVIOUS REVIEW (95%CI were added in relation to the mean values)- Please also include the 95% confidence intervals for each correlation and explain how the magnitude of each correlation was interpreted. In addition, due to the multiple correlations you must correct each p value for familywise error rates. This must them be updated throughout the results and discussion. Please also provide the threshold by which the r values are interoretted.
Confidence intervals were added in Table 3. In tables 5 and 6 the numer of multiple correlations of one variable with other is equal to 7 and 4 respectively. That means that the probability of avoiding Type I error are 0.698 and 0.815 respectively. These values do not differ much from the probability of avoiding an error for single measurement i.e. ( 1-α= 0.95). That is the reason for omitting the adjusting of p according to Boniferroni. Confidence intervals in correlations were not added due to sample size